# Discord in Concordance Cosmology and Anomalously Massive Early Galaxies

Stacy S. McGaugh

Department of Astronomy, Case Western Reserve University, 10900 Euclid Avenue, Cleveland, OH 44106, USA; stacy.mcgaugh@case.edu

**Abstract:** Cosmological parameters are constrained by a wide variety of observations. We examine the concordance diagram for modern measurements of the Hubble constant, the shape parameter from the large-scale structure, the cluster baryon fraction, and the age of the universe, all from non-CMB data. There is good agreement for $H_0 = 73.24 \pm 0.38 \, \text{km s}^{-1} \, \text{Mpc}^{-1}$ and $\Omega_m = 0.237 \pm 0.015$. This concordance value is indistinguishable from the WMAP3 cosmology but is not consistent with that of Planck: there is a tension in $\Omega_m$ as well as $H_0$. These tensions have emerged as progressively higher multipoles have been incorporated into CMB fits. This temporal evolution is suggestive of a systematic effect in the analysis of CMB data at fine angular scales and may be related to the observation of unexpectedly massive galaxies at high redshift. These are overabundant relative to $\Lambda$CDM predictions by an order of magnitude at $z > 7$. Such massive objects are anomalous and could cause gravitational lensing of the surface of last scattering in excess of the standard calculation made in CMB fits, potentially skewing the best-fit cosmological parameters and contributing to the Hubble tension.

**Keywords:** ingbackground radiation; cosmic; early universe; Galaxy formation; gravitational lensing; Hubble constant



## 1. Concordance Cosmology

The concordance cosmology ($\Lambda$CDM) emerged in the 1990s thanks to observational advances that constrained a broad variety of cosmological parameters [1,2]. These led to the surprising recognition that the mass density was less than the critical density [3], in conflict with the then-standard SCDM cosmology and Inflationary theory. Retaining the latter implied a second surprise in the form of the cosmological constant such that $\Omega_m + \Omega_\Lambda = 1$, which in turn predicted that the expansion rate of the universe would currently be *accelerating* [4]. This unlikely-seeming prediction was subsequently observed in the Hubble diagram of Type Ia SN [5,6]. This picture was further corroborated by the location of the first peak of the acoustic power spectrum of the CMB being consistent with a flat ($\Omega_k = 0$) Robertson–Walker geometry [7,8].

The range of concordance parameters narrowed with the completion of the Hubble Space Telescope Key Project to Measure the Hubble Constant [9], and a 'vanilla' set of $\Lambda$CDM parameters emerged with[1] $h = 0.7$ and $\Omega_m = 0.3$ [10] some twenty years ago. Since that time, a tension [11–13] has emerged between direct measurements of the Hubble constant [9,14–20] and that obtained from fits to the acoustic power spectrum of the cosmic microwave background obtained by the Planck mission [21]. This tension was not present in earlier data from WMAP [22]. Indeed, one of the most persuasive arguments in favor of vanilla $\Lambda$CDM was the concordance of WMAP cosmological parameters with other observations, especially $H_0$. Now that a tension has emerged, it is natural to ask when and where things went amiss.

To this end, it is interesting to update the $\Omega_m$-$H_0$ concordance diagram that was pivotal in establishing $\Lambda$CDM in the first place [2]. Figure 1 shows modern values of the

same constraints used in Ref. [2] nearly three decades ago. These include the SH0ES direct measurement of the Hubble constant $H_0 = 73.04 \pm 1.04$ km s$^{-1}$ Mpc$^{-1}$ [19], the shape parameter $\Omega_m h = 0.168 \pm 0.016$ from the large-scale structure [23], the baryon fraction of clusters of galaxies $\Omega_m h^{1/2} = 0.221 \pm 0.031$ [24], and the age of the globular cluster M92, $13.80 \pm 0.75$ Gyr [25]. For the age constraint, we assume a flat geometry as an open universe without a cosmological constant, as it would be too young to have a concordance region. For simplicity, we also assume the age of the universe is indistinguishable from the age of the globular cluster, which presumably formed very early and surely within the first 750 Myr (the uncertainty in its age).

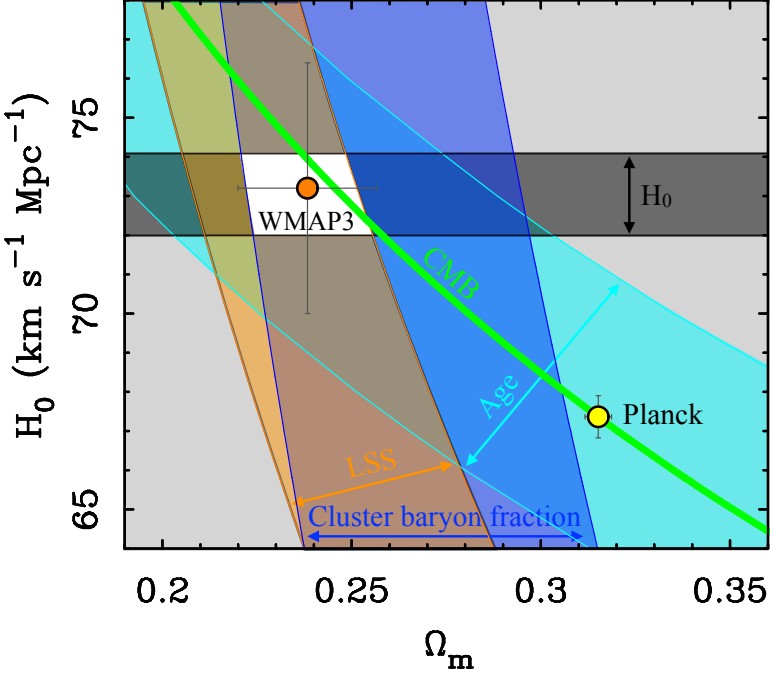

**Figure 1.** The concordance region (white space) in $\Omega_m$-$H_0$ space [2] where the allowed regions (colored bands) of many constraints intersect. Illustrated constraints include a direct measurement of the Hubble constant [19] [black], the age of a flat universe containing the globular cluster M92 [25] [light blue], the cluster baryon fraction [24] [dark blue], and the shape parameter from the large-scale structure [23] [orange]. This modern rendition of the concordance region is consistent with a WMAP3 cosmology [22] [orange point] but not with that of Planck [21] [yellow point]. The green line represents the covariance of CMB fits, $\Omega_m h^3 = 0.09633 \pm 0.00030$ [21].

The data in Figure 1 tell very much the same story as it has for the entirety of this century. Taking these observations at face value, we recover a concordance cosmology with $\Omega_m = 0.235 \pm 0.015$ and $h = 0.7304 \pm 0.0104$. This is completely consistent with the WMAP3 [22] cosmology ($\Omega_m = 0.241 \pm 0.034$, $h = 0.732 \pm 0.032$), but less so with the later results.

## 2. Variations on the Concordance Diagram

The most accurate constraint in Figure 1 is the direct measurement of the Hubble constant from SH0ES [19], so it is worth considering what happens if we adopt other values. There are several groups that have independently achieved $\sim 1\%$ precision with random errors $< 1$ km s$^{-1}$ Mpc$^{-1}$ [14,19,20]. Considering only the random error, the CosmicFlows-4 measurement $H_0 = 74.6 \pm 0.8$ [20] is consistent with the SH0ES value ($H_0 = 73.04 \pm 1.04$ km s$^{-1}$ Mpc$^{-1}$), differing by only $1.2\,\sigma$, while the CCHP value of $H_0 = 69.8 \pm 0.8$ km s$^{-1}$ Mpc$^{-1}$ [14] is marginally inconsistent at the $\sim 2.5\,\sigma$ level.

Taking these measurements of $H_0$ and their statistical uncertainties at face value, we can reconstruct the concordance diagram for each (Figure 2). Combining the CCHP

$H_0 = 69.8 \pm 0.8 \, \mathrm{km \, s^{-1} \, Mpc^{-1}}$ [14] with the other constraints leads to a concordance region with $\Omega_m = 0.247 \pm 0.017$. Doing the same for the CosmicFlows-4 $H_0 = 74.6 \pm 0.8$ [20] gives $\Omega_m = 0.233 \pm 0.013$. Treating these as the practical range of measured Hubble constants, the corresponding range of mass density is $0.22 \leq \Omega_m \leq 0.264$. This range is consistent with local estimates of the mass density [26,27] and with WMAP cosmologies, but not with the $\Omega_m = 0.315 \pm 0.007$ of the Planck cosmology [21].

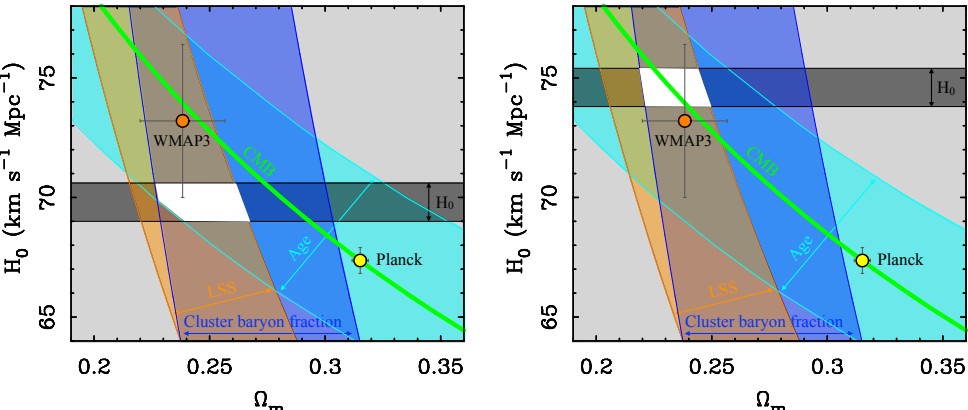

**Figure 2.** The same as Figure 1 but a lower (**left**) and higher (**right**) choice of $H_0$ to illustrate the range of systematic uncertainty in direct determinations of the Hubble constant. Neither choice reconciles the concordance of local cosmological constraints with the Planck cosmology.

Figure 3 summarizes the status of the Hubble tension as of this writing. Local measurements range from 69 to 75 $\mathrm{km \, s^{-1} \, Mpc^{-1}}$ obtained by a variety of methods and calibrations. This subject is the focus of much research with frequent updates to essentially the same result, e.g., Freedman et al. [14] and Freedman [28], so we restrict Figure 3 to a single entry from each distinct group. We highlight the most precise measurements with statistical uncertainties $\leq 3 \, \mathrm{km \, s^{-1} \, Mpc^{-1}}$. The weighted mean of these values gives $H_0 = 73.24 \pm 0.38 \, \mathrm{km \, s^{-1} \, Mpc^{-1}}$. The range of mass density consistent with this and the other constraints in the concordance diagram is $\Omega_m = 0.237 \pm 0.015$. This precise estimate from the concordance of local observations is indistinguishable from the WMAP3 [22] cosmology.

The above assessment is both straightforward and naive, as it ignores systematic errors, which are difficult to quantify. The history of the distance scale predisposes us to suspect a systematic error somewhere in the calibration of the distance ladder. This view is encouraged by the apparent tension between Cepheid [19] and TRGB [14] calibrations. However, there is no tension between these two calibrators in other analyses [17,18,20], so this may not be a difference between Cepheid and TRGB methods so much as it is between the distinct TRGB methods employed by different groups [29,30].

An assessment of systematics is built into the SH0ES error bar [19], so by their assessment, the tension with Planck remains highly significant ($5\sigma$). Independent assessments using the Tully–Fisher relation [31] consistently find $H_0$ consistent with the SH0ES value or even a tad higher [12,20], which goes in the wrong direction to achieve consistency with Planck. Indeed, application of the baryonic Tully–Fisher relation [32] excludes $H_0 < 70.5 \, \mathrm{km \, s^{-1} \, Mpc^{-1}}$ at 95% c.l. [17]; the asymmetric probability distribution skews to higher rather than lower values, so $H_0 < 68 \, \mathrm{km \, s^{-1} \, Mpc^{-1}}$ is practically excluded, barring a systematic error in the calibration of both Cepheids and the TRGB method. The plausible amplitude of such a systematic calibration uncertainty is suggested to be $\pm 1.3 \, \mathrm{km \, s^{-1} \, Mpc^{-1}}$ by the CCHP [33]. This almost reconciles the low estimate of $H_0$ from CCHP with Planck: the tension is only $1.6 \, \sigma$ *if* both random and systematic uncertainties are at their maximum and combine to go in the same direction. Setting aside the eternal question of how much to believe the calibrations and the assessments of systematic uncertainty [12,13], there are also geometric methods [15,16] that are independent of stellar

calibrators. These also favor $H_0$ in the low to mid-70s (Figure 3). Consequently, the essential answer has not changed since the completion of the Hubble Space Telescope Key Project to Measure the Hubble Constant [9]. Progress has been made in accuracy, which has improved to the point that it is difficult to sustain the presumption that there must be a systematic error in the distance scale [12].

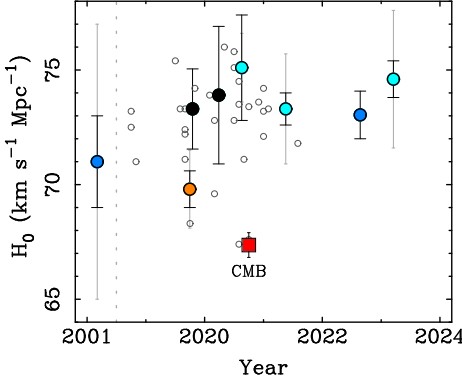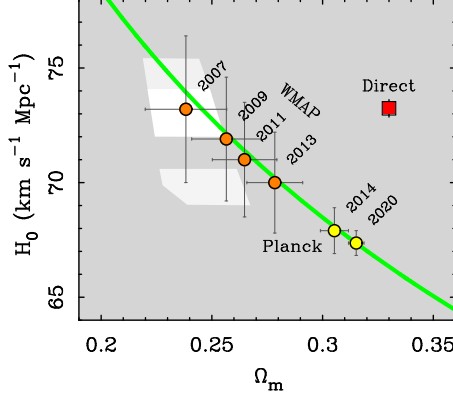

**Figure 3.** Recent direct measurements of $H_0$ [11] over time (**left panel**) including the earlier HST key project [far left] [9]. Measurements with statistical uncertainties $\leq 3 \, \mathrm{km \, s^{-1} \, Mpc^{-1}}$ from independent groups are color coded by calibrator: geometry [black] [15,16], Cepheids [blue] [9,19], TRGB [orange] [14], or a combination [cyan] [17,18,20]. Statistical and systematic uncertainties are shown as dark and light error bars, respectively; these are omitted for less accurate data (small open circles). Determinations of $H_0$ from WMAP [orange] [22,34–36] and Planck [yellow] [21,37] covary with the mass density as $\Omega_m h^3 = 0.09633 \pm 0.00030$ [green line] [21] and have decreased steadily over time (**right panel**). For comparison, the red square shows the latest CMB value in the left panel and the mean of the accurate direct determinations in the right panel (at arbitrary $\Omega_m$). The concordance region from Figure 1 is shown as the open region in the right panel; alternate concordance regions from Figure 2 are shown as light gray. The remaining region (dark gray) is excluded by those data.

Tension in $\Lambda$CDM is not restricted to the Hubble constant; it also appears in the mass density. Indeed, it appears using modern measurements of the same observational constraints that established the concordance cosmology in the first place [2]. A major reason for the persistence of $\Lambda$CDM was the excellent agreement between early WMAP cosmologies and the wealth of previous constraints that all indicated the same region of concordance in Figure 1.

A similar success does not extend to Planck cosmologies. This is true for any choice of locally measured Hubble constant. While obvious for the higher estimates of $H_0$, even the lowest value from the CCHP is not really in agreement with Planck so much as it is not in strong tension with it. However, if we also consider the mass density implied in Figure 2, then the tension is magnified: not only is $H_0$ too big, but $\Omega_m$ is too low. Given the different dependencies on $h$ of the various constraints, there is no prospect of regaining concordance. Even if a local measurement gave exactly the same Hubble constant as Planck, the concordance region would remain at lower $\Omega_m \approx 0.27$ rather than 0.315, simply transferring the tension in $H_0$ to one in $\Omega_m$. Indeed, the concordance window for any $H_0 < 72 \, \mathrm{km \, s^{-1} \, Mpc^{-1}}$ fails to contain the locus of $\Omega_m h^3$ demanded by CMB data (the green line in Figure 3); there is no FLRW universe that satisfies all the illustrated constraints.

Faced with the choice between abandoning the FLRW cosmology and disbelieving some aspect of the data, most people choose the latter. However, it is becoming difficult to do so, as other tensions do exist. For example, there is a persistent tension in the power spectrum normalization $\sigma_8$ [38–40]. There is also a significant but oft-neglected tension in the baryon density between deuterium [41] and lithium [42]. The deuterium value agrees with CMB fits, so the general perception seems to be that lithium must be in systematic[2]

error. This is equivalent to assuming the Planck estimate of $H_0$ is correct and all local measurements must inevitably be in error.

The most consequential constraint considered here, after direct measurements of $H_0$, is that on the product $\Omega_m h$ of the large-scale structure from the 2dF Galaxy Redshift Survey [45]. The 2dF value, $\Omega_m h = 0.168 \pm 0.016$ [23], limits how large a mass density is allowed, and precludes concordance with the higher value of Planck ($\Omega_m h = 0.212 \pm 0.005$). Ironically, the preliminary result of the same survey initially gave a more consistent result, $\Omega_m h = 0.20 \pm 0.03$ [46], but the answer changed as the larger survey volume was analyzed [23]. The difference between the initial and final result of the 2dF survey gives cause to ponder the extent to which the structure of the local universe is consistent with $\Lambda$CDM [47–49], with some even calling into question the basic cosmological premise of isotropy [50–54].

At this juncture, there are so many measurements that it is possible to find indications of tensions in any cosmological parameter. The trick is to assess the credibility of each while avoiding confirmation bias. It is also important to bear in mind the possibility that the concordance window is indeed closed and there is no viable FLRW universe. It is conceivable that the strange parameters of $\Lambda$CDM are merely the best FLRW approximation to some deeper theory [55–60].

### 3. Covariance and the Hubble Tension

The best fit to the Planck CMB data, $H_0 = 67.36 \pm 0.54$ km s$^{-1}$ Mpc$^{-1}$ [21], clearly differs from the bulk of the accurate "local" determinations [9,14–20]. The weighted mean of these values differs from the Planck value by nearly $9\sigma$, so formally, the tension is real. Though systematic uncertainties are inevitable at some level, that level is around $\pm 1.3$ km s$^{-1}$ Mpc$^{-1}$ [33], not the $\sim 6$ km s$^{-1}$ Mpc$^{-1}$ that would be required to reconcile the two: the tension remains $4.5\,\sigma$ for plausible systematic uncertainties. In other words, as the precision of local distance scale measurements have approached 1%, their accuracy has reached $\sim$2%, but we need a systematic of 9% to make the problem go away.

Local measurements of the distance scale and fits to the CMB measure completely different things. The local measurements are all some realization of the traditional Hubble program in which distances and redshifts to nearby galaxies are measured, and the slope of the velocity–distance relation is obtained. This is an empirical procedure. In contrast, fits to the power spectrum of the CMB at $z = 1090$ are model ($\Lambda$CDM) dependent, and the results are sensitive to all of the cosmic parameters simultaneously with inevitable covariance. The strongest covariances affecting the Hubble parameter is with the mass density. These covary in CMB fits approximately as $\Omega_m h^3 = 0.09633 \pm 0.00030$ [21].

The covariance of $H_0$ with $\Omega_m$ is apparent in Figure 3. There is also a correlation with time: the best-fit combination has marched steadily along the trench of constant $\Omega_m h^3$ from WMAP3 with $H_0 = 73.2 \pm 3.2$ km s$^{-1}$ Mpc$^{-1}$ [22] to Planck with $H_0 = 67.36 \pm 0.54$ km s$^{-1}$ Mpc$^{-1}$ [21]. It is the CMB result that has progressively deviated from the concordance $\Lambda$CDM region established around the turn of the millennium, not local measurements.

The temporal progression of the best-fit values of $H_0$ and $\Omega_m$ is a flag that suggests something systematic in the CMB analyses rather than traditional Hubble constant determinations. The latter scatter approximately as expected: the rms variation in the accurate data in Figure 3 is 1.78 km s$^{-1}$ Mpc$^{-1}$, while the size of the statistical uncertainties anticipate that it should be 1.55 km s$^{-1}$ Mpc$^{-1}$. This implies that systematic uncertainties are not contributing much to the scatter, though one must always be cautious of a calibration issue that would translate all results without affecting the scatter [33].

The CMB data from independent experiments are in good agreement where they overlap, so the temporal progression is not a measurement error. What has changed over time is the availability of data on ever finer scales, tracing the power spectrum to higher multipoles $\ell$. Something subtle seems to be tipping the covariance of parameters to favor lower $H_0$ along the trench of constant $\Omega_m h^3$ as higher $\ell$ have been incorporated into the fits.

## 4. Massive Galaxies at High Redshift

An important recent development is the observation of unexpectedly massive galaxies at high redshift. This may seem unrelated at first, but CMB data have reached the level of precision where the subtle effects of gravitational lensing by masses intervening between ourselves and the surface of last scattering cannot be ignored. If massive galaxies assembled anomalously early, then gravitational lensing by these objects may have a systematic impact on CMB fits.

Galaxies are expected to assemble gradually via the hierarchical merger of many progenitors in $\Lambda$CDM. This process takes a long time. A typical massive galaxy is predicted to have assembled half its stars around $z \approx 0.7$ when the universe is $\sim$7 Gyr old, roughly half its current age [61,62]. In contrast, there are many galaxies that are observed to be massive and already quiescent at $z \approx 3$ [63–66] when the universe is only $\sim$2 Gyr old. These appear to have formed half of their stars in the first gigayear ($z \gtrsim 6$), and are consistent with the traditional picture of a monolithic early-type galaxy that formed early ($z \geq 10$) and followed an approximately exponential star formation history with a short ($\sim$1 Gyr) e-folding time [64]. This comes as a surprise[3] to the predictions of hierarchical structure formation in $\Lambda$CDM [70,71].

Though highlighted by recent JWST observations at $z \approx 10$ [72–77], the discrepancy is already apparent by $z \approx 3$ in earlier observations [63,64]. Figure 4 illustrates the situation shortly before the launch of JWST. The observed luminosity function of galaxies broadly resembles that predicted, but there is a systematic excess of bright, massive galaxies. This persists over all redshifts $z > 3$ and holds in all environments, both in clusters and in the field [64].

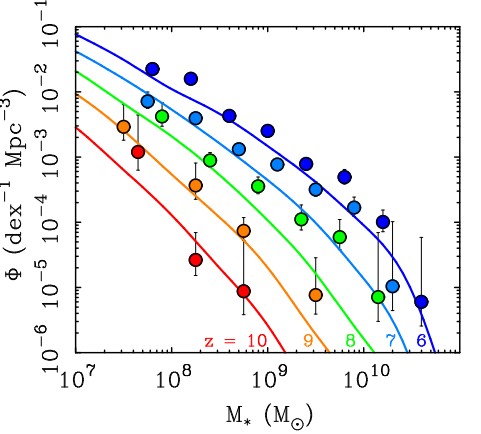 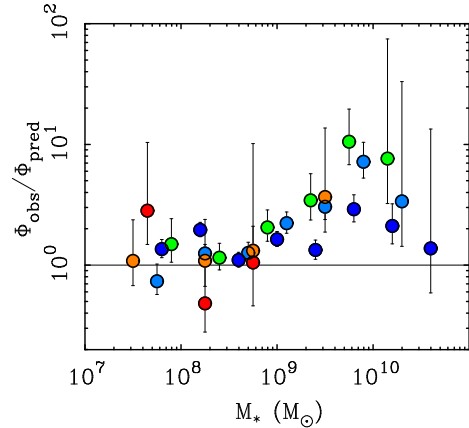

**Figure 4.** The number density $\Phi$ of galaxies as a function of their stellar mass $M_*$, color coded by redshift with $z = 6, 7, 8, 9, 10$ in dark blue, light blue, green, orange, and red, respectively. The left panel shows predicted stellar mass functions [lines] [78] with the corresponding data [circles] [79]. The right panel shows the ratio of the observed-to-predicted density of galaxies. There is a clear excess of massive galaxies at high redshifts.

The 'impossibly early galaxy' problem [63] has become more severe with new observations from JWST [80], extending to $z > 9$ (Figure 5). Over this redshift range, the luminosity function is predicted to evolve rapidly [81]. There is a good physical reason for this, as this is the epoch of hierarchical mass assembly in $\Lambda$CDM when fragmentary protogalaxies should first be forming before subsequently assembling into more massive galaxies at much later times. There is not sufficient time in $\Lambda$CDM to assemble the observed mass into single objects [70,71], hence the ragged luminosity functions in the highest redshift [4] bins.

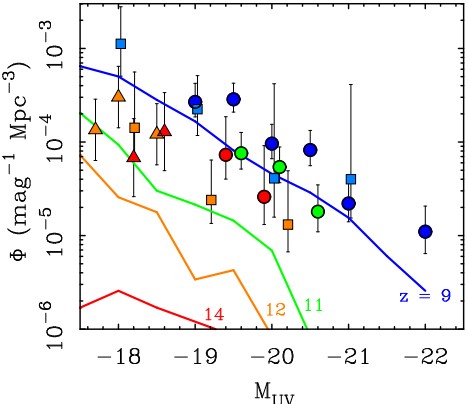
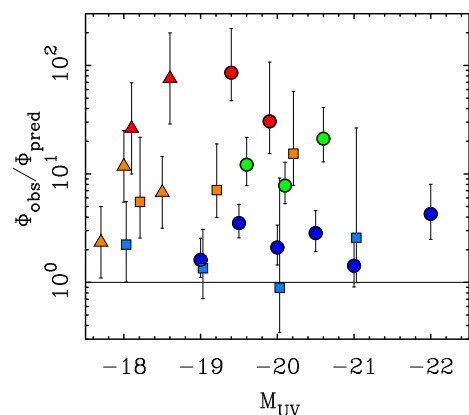

**Figure 5.** The number density of galaxies as a function of their rest-frame ultraviolet absolute magnitude observed by JWST, a proxy for stellar mass at high redshift. The left panel shows predicted luminosity functions [lines] [81], color coded by redshift: blue, green, orange, red for $z = 9, 11, 12, 14$, respectively.Data in the corresponding redshift bins are shown as squares [82], circles [83], and triangles [84]. The right panel shows the ratio of the observed-to-predicted density of galaxies. The observed luminosity function barely evolves, in contrast to the prediction of substantial evolution as the first dark matter halos assemble. There is a large excess of bright galaxies at the highest redshifts observed.

These JWST results are new, and will no doubt be debated for some time. Valid concerns can be raised over the veracity of photometric redshifts and the relation of ultraviolet starlight to dark matter halo mass. However, the anomalous presence of early massive galaxies found by JWST (Figure 5) corroborates previous work (Figure 4) for which spectroscopic redshifts are available and for which the assessment of stellar mass is more robust. It is therefore hard to avoid the conclusion that galaxies grew too big too fast.

## 5. Gravitational Lensing of the CMB

The presence of massive galaxies in the early universe is an anomaly to the $\Lambda$CDM structure formation paradigm. They should not be there [70,71]. The existence of such galaxies violates the assumptions that underpin fits to the acoustic power spectrum of the CMB, so it may lead to systematic errors in the assessment of cosmological parameters based on those fits. In particular, the calculation of gravitational lensing would be impacted [85].

Gravitational lensing of the surface of last scattering by intervening masses is an effect that becomes important at high $\ell$, beginning to impact the fit for $\ell > 800$. Intriguingly, restricting the fit of the Planck data to $\ell < 800$ returns a higher Hubble constant [21] consistent with that found by WMAP [36]. Consequently, concordance with local data is maintained for low-$\ell$ data; the discrepancy has only appeared as high-$\ell$ data have been incorporated into the fits. So perhaps there is a systematic effect due to the gravitational lensing of the surface of last scattering.

We have seen above that massive galaxies appear anomalously early in the history of the universe. This implies a stronger lensing effect than expected. In order to compute the effects of lensing on the observed CMB, lensing potentials are extrapolated forward in time by assuming linear growth of the perturbations observed in the CMB. The JWST data imply that this assumption is violated, but it is hardwired into the calculations used to fit CMB data. This implies that the computation of lensing underestimates the real effect by assuming a growth rate that is less than observed.

High mass galaxies at high redshift are anomalous, so we do not know how to calculate the lensing potentials they represent. We can, however, use CAMB [86] to compute the CMB power spectrum to see how the predicted power varies in the absence of this effect. We do this in Figure 6, which compares the Planck best fit with a model that has $H_0 = 73 \, \mathrm{km \, s^{-1} \, Mpc^{-1}}$. We scale the mass density of this model to maintain consistency

with the constraint $\Omega_m h^3 = 0.09633$ [21]. We hold the mass density of baryons and neutrinos constant, scaling that of cold dark matter to $\Omega_{CDM} h^2 = 0.109$ for a total mass density of $\Omega_m = 0.248$, consistent with the concordance region in Figure 1. This gives an acoustic power spectrum that is only distinguishable from the best-fit version upon close scrutiny (Figure 6), at a level where the effect of gravitational lensing is perceptible.

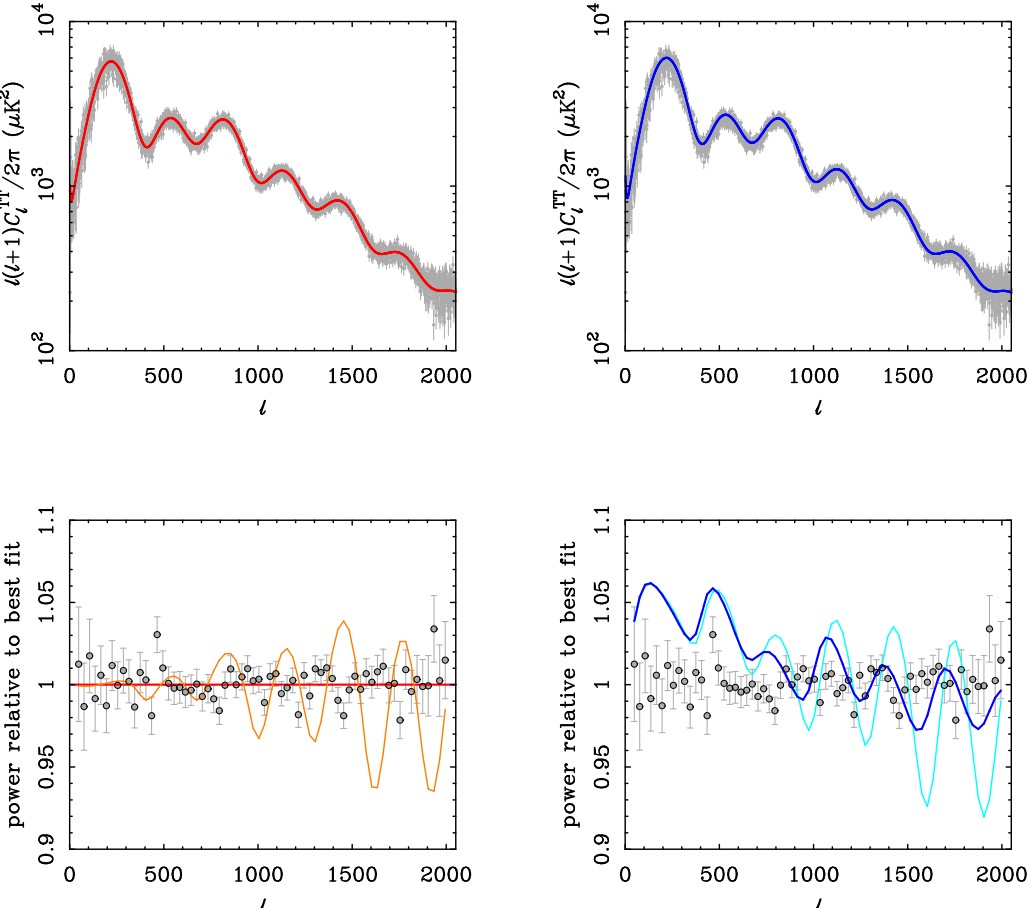

**Figure 6.** (**Top**): unbinned Planck data [87] with the best-fit power spectrum (**left**) and a model with $H_0 = 73 \text{ km s}^{-1} \text{ Mpc}^{-1}$ (**right**) with $\Omega_{CDM}$ scaled to keep $\Omega_m h^3 = 0.09633$ constant. (**Bottom**): the ratio of binned data and model to the best fit. The darker (red/blue) line is the model including the expected effects of lensing; the lighter (orange/cyan) line is the power spectrum emergent from the surface of last scattering before lensing. If there is an anomalous source of lensing from massive early galaxies, then the mapping from emergent-to-observed power spectrum will be miscalculated and the best-fit cosmological parameters may be in error.

The lensing calculation, though subtle, plays an important role in the fit. Lensing blurs the surface of last scattering, suppressing some of the oscillations that were emergent at the time of recombination, as illustrated by the lighter lines in the lower panels of Figure 6. If the predicted amount of lensing is wrong because of the anomalously early appearance of massive galaxies, the resulting fit will be systematically in error.

The power spectrum of the model with $H_0 = 73 \text{ km s}^{-1} \text{ Mpc}^{-1}$ is offset and tilted from the Planck best fit. The offset at $\ell < 800$ is mostly caused by covariance with the optical depth to the surface of last scattering, which shifts the amplitude. WMAP found $\tau = 0.089 \pm 0.014$ [36] as rather higher than the best fit of Planck, $\tau = 0.0544 \pm 0.0073$ [21]. This is about what it takes to shift the model into agreement with the data at low $\ell$, which is part of why WMAP found a higher best-fit optical depth. There are, of course, other covariances, especially with the amplitude of the power spectrum, $\sigma_8$, which itself is in some tension with local measurements [38–40].

In addition to blurring the surface of last scattering, gravitational lensing also adds a source of small scale power to the observed fluctuations that are not present in the power spectrum emergent from the surface of last scattering. This will add poorly correlated power at high $\ell$. If there is an excess of such power from anomalously early galaxies, it will tilt the power spectrum in the sense seen in Figure 6. That is, if one starts with a model like that with $H_0 = 73 \,\mathrm{km\,s^{-1}\,Mpc^{-1}}$, power will be added at high $\ell$ beyond what can be explained by the model. If this happens but we are unaware of it, the fitting procedure will indulge the covariance of all the cosmological parameters until a fit is found. Driving $H_0$ down and $\Omega_m$ up goes in the right direction to do this. It is therefore conceivable that the Planck best-fit $H_0$ is in systematic error due to anomalous gravitational lensing.

## 6. Conclusions

We have made an updated assessment of the concordance diagram that gave rise to $\Lambda$CDM [2]. Using modern data, we find $H_0 = 73.24 \pm 0.38 \,\mathrm{km\,s^{-1}\,Mpc^{-1}}$ and $\Omega_m = 0.237 \pm 0.015$. These values are essentially unchanged from twenty years ago and are in good agreement with the WMAP3 cosmology. They are not consistent with the Planck cosmology: the tension is present in the mass density as well as the Hubble constant.

Cosmological parameters obtained from fits to the acoustic power spectrum of the cosmic microwave background have gradually moved away from the concordance region specified by independent constraints, moving steadily along the trench of constant $\Omega_m h^3$ to lower $H_0$ and higher $\Omega_m$ leading to the current tension with the locally measured value of the Hubble constant. This migration correlates with the inclusion of higher multipoles in the fits, for which the effects of gravitational lensing are important. The appearance of anomalously massive galaxies in the early universe as indicated by JWST (and other) observations suggests that the impact of gravitational lensing on the CMB may be underestimated. It is conceivable that this is a contributing factor to the temporal migration of the best-fit cosmological parameters and the resulting Hubble tension. If so, it may be the CMB value of $H_0$ that is systematically in error rather than local determinations.

Massive galaxies at high redshift are anomalous in $\Lambda$CDM and would require new physics to explain. We have refrained from speculating on what such new physics might be, but note that it was predicted in advance by MOND [43,67]. In addition to specifying a particular hypothesis for the new physics, it would further be necessary to incorporate the predicted growth rate into a Boltzmann code in order to calculate the lensing effect on the surface of last scattering. Such a calculation is beyond the scope of this work.

**Funding:** This research received no external funding.

**Conflicts of Interest:** The author declares no conflicts of interest.

## Abbreviations

The following abbreviations are used in this manuscript:

| | |
|---|---|
| CCHP | Carnegie-Chicago Hubble Program |
| CMB | Cosmic Microwave Background |
| JWST | James Webb Space Telescope |
| $\Lambda$CDM | Lambda Cold Dark Matter |
| SCDM | Standard Cold Dark Matter |
| SH0ES | Supernovae and $H_0$ for the Equation of State of dark energy |
| WMAP | Wilkinson Microwave Anisotropy Probe |

## Notes

1     $h = H_0/(100 \, \text{km} \, \text{s}^{-1} \, \text{Mpc}^{-1})$.

2     There was no discrepancy prior to the appearance of CMB constraints [43]. Since then, lithium estimates have remain unchanged while deuterium changed suddenly to come into concordance [44].

3     The formation of massive galaxies at $z \geq 10$ was predicted in advance by Bob Sanders in the context of MOND [67], in which case the expansion history of the early universe may differ from $\Lambda$CDM [68]. The problems posed by massive early galaxies may also be relieved if the universe is much older than generally thought [69].

4     A luminosity density of $\sim 7 \times 10^{-6}$ mag.$^{-1}$ Mpc$^{-3}$ is estimated at $z \approx 16$ [82]. This point is omitted from Figure 5 because the corresponding prediction is not a number: galaxies bright enough to observe should not yet exist.

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
