# Peer review of "Discord in Concordance Cosmology and Anomalously Massive Early Galaxies"

_universe, doi:10.3390/universe10010048_

Round 1

Reviewer 1 Report

Comments and Suggestions for Authors

This article discusses the important issues of the accuracy of the Hubble constant and other cosmological parameters. It may be accepted in the present form. 

Strength: 

The proposed hypothesis  that the impact of gravitational lensing on the CMB may be underestimated is interesting. Massive galaxies at high redshift is anomalous. New physics is required to explain this phenomen. The author spequlates what it would be.

Weakness:

The reasons for the discrepancies between observational estimates of cosmological parameters and their errors are only briefly discussed. 

Interesting perspectives to solving the problem are provided by a closer look at the geometry of space structures. Please, take a look at the paper by Neuzil M.K., MNRAS 494, 2600 (2020) wich considers the overdensity of giant galaxies in the Local universe.

Minor ethical concern:

An article with the same title was published by S.S. McGaugh:  

 2023RNAAS...7...20M, DOI: 10.3847/2515-5172/acba9a

Author Response

I thank the referee for his/her comments, and attention at this time of year.

I am apparently not very creative with titles. I considered several, and somehow worked my way to one very similar to one I had used before. It has been changed to better represent the emphasis on the concordance region in this manuscript.

I have added another figure to illustrate the uncertainties, with an expended discussion thereof. 

I thank the referee for pointing out the over-density of local volume galaxies, and agree it is important to make the connection to the persistence of this phenomenon at low redshift, as Peebles and Nusser have also noted, so I add these references. 

Reviewer 2 Report

Comments and Suggestions for Authors

This is an interesting manuscript given the Webb results.

However, in the concluding section it could be useful to compare with the alternative or speculative CCC propososal of  Roger Penrose (2006). "Before the Big Bang: An Outrageous New Perspective and its Implications for Particle Physics" (PDF)Proceedings of the EPAC 2006, Edinburgh, Scotland: 2759–2762. Bibcode:2006epac.conf.2759R.

as well as with the recent paper of R Gupta, JWST early Universe observations and ΛCDM cosmology, Monthly Notices of the Royal Astronomical Society (2023). DOI: 10.1093/mnras/stad2032

Author Response

I thank the referee for his/her attention, particularly at this time of year. While I have myself refrained from speculating too much about the nature of new physics that might drive the presence of early galaxies, the referee points out papers that are indicative of such. I have noted them in the revised submission. 

Reviewer 3 Report

Comments and Suggestions for Authors

The author discuss a tension between the expansion rate of the universe estimated from direct measurements and that from fits of CMB. In the paper it is suggested that this discrepancy is from gravitational lensing effect from high red shift galaxies motivated by recent astrophysical observations. I think the discussion is interesting to show possibility to explain the tension. So I think the manuscript can be published. A few comments are as follows:

*It would be good to explain definition of M_{UV} (ultraviolet absolute magnitude) used in Fig.4

*At line 84 of page 3, word "the completion of" is duplicated

Author Response

I thank the reviewer for his/her comments, and for their punctuality at this time of year. I have removed the duplicate text that was spotted (sharp eyes!) and expanded a bit the description of the UV absolute magnitude. This has become standard in the community working on high-z JWST data but is rather obscure to the rest of us. The importance of this quantity is as a normalized, rest-frame luminosity that is used as a proxy for stellar mass.

Round 2

Reviewer 2 Report

Comments and Suggestions for Authors
  1. The autor may also cite the follow-up on CCC in FP: 
  2.   On the Gravitization of Quantum Mechanics 1: Quantum State Reduction.Roger Penrose - 2014 - Foundations of Physics 44 (5):557-575.

Author Response

OK, I am happy to add these additional references.